# Clean-Action Backdoor Attacks on Vision-Language-Action Models via Sequential Error Exploitation

## Abstract

Vision-Language-Action (VLA) models have emerged as a popular method for general-purpose embodied AI, enabling robots to interpret multimodal inputs and generate temporally coherent actions. Popular imitation learning methods, including diffusion-based and autoregressive approaches, typically rely on human-collected demonstrations, which often contain small execution errors such as pauses or irregular motions even when consisting only of successful trajectories. Because decision-making in robotics is sequential, even small errors can compound over time, eventually leading to task failure. In this work, we exploit this property to introduce a new class of clean-action backdoor attacks, which require only partial poisoning of demonstration trajectories while preserving overall roll-outs and apparent task success. Unlike conventional backdoors, our approach is more difficult to detect, since it conceals malicious behaviors within natural error patterns rather than obvious trajectory alterations. We validate our method by backdooring the $\pi_0$ model and testing on the LIBERO benchmark, where it achieves consistently high attack success rates while evading standard detection and remaining effective under clean-data fine-tuning. These findings highlight the urgent need for VLA-specific defenses that address sequential vulnerabilities in embodied AI systems.

## 1 Introduction

Vision-Language-Action (VLA) models fuse large vision-language backbones with action decoders to produce end-to-end policies that interpret visual scenes, follow natural-language instructions, and emit temporally coherent control sequences. Recent representative models such as Discrete Diffusion VLA Liang et al. (2025), $\pi_0$ Black et al. (2024), OpenVLA Kim et al. (2024) and RT-2 Zitkovich et al. (2023) have achieved strong generalization on various robot manipulation benchmarks Liu et al. (2023); Chen et al. (2025) by leveraging scale and diverse robot datasets Khazatsky et al. (2024); O'Neill et al. (2024).

This strong dependence on large, diverse demonstration corpora raises a practical concern: when fine-tuning relies on third-party or semi-trusted datasets, how can users reliably distinguish harmful data from useful data? A compromised dataset not only degrades learned performance but can also introduce security risks, most notably backdoors that cause VLA models to behave incorrectly. As VLA models expand to broader environments and tasks, and depend on increasingly large datasets, this vulnerability becomes both more realistic and more pressing to study.

Traditional backdoor and poisoning research demonstrates that a small number of carefully crafted training examples can implant persistent failure modes in learned models Gu et al. (2017); Shafahi et al. (2018); Turner et al. (2019), and recent multimodal and encoder-targeted attacks show that representation-level manipulations can transfer across modalities Liang et al. (2024); Walmer et al. (2022); Yang et al. (2023). However, many of these techniques are difficult to apply naively in VLA settings because robotic demonstrations come with strong simulation-grounded filtering (dynamics, kinematics, rendering), which make common data poisoning easy to detect. Conversely, clean-label methods from image classification Turner et al. (2018); Shafahi et al. (2018) are designed to bypass dataset filtering and could be introduced to VLA settings. However, they often introduce subtle

degradation that hurts task-relevant metrics, such as time-to-completion and energy consumption, that are as important as success rate.

To address these challenges, we propose a class of *clean-action* backdoor attacks that balance the stealthiness towards data filtering, benign performance and attacking performance. Our key observation is that: during sequential decision-making, small and natural errors in demonstrations, such as pauses and noises, can get much worse when a learned policy makes mistakes over time. VLA policies normally predict a short sequence (an action chunk) and then execute the first few actions from that chunk before replanning. If a perturbation lasts across the portion of the chunk that the agent actually executes, the model cannot correct it until the next replanning step. Repeated or correlated perturbations therefore accumulate and push the agent farther from the desired trajectory. Due to this observation, backdoor attacks targeting at task failures no longer require a significant mistake in demonstrations and this indicates the potential of stealthy backdoor attacks under dataset filtering.

Our main contributions are as follows:

- **Practical threat model.** We identify a simple and effective threat model by considering a natural dataset filtering scheme that is able to prevent most backdoor attacks using data poisoning. It poses a higher requirement of stealthiness for the adversaries.

- **Clean-action backdoor attack.** We propose a new backdoor attack method for VLA models which consists of a poisoning protocol and a data augumentaion recipe if needed. This kind of attack doesn't introduce significant action modifications so that it can bypass the dataset filtering.

- **Empirical and analytical study.** We test the attack with $\pi_0$ model on LIBERO benchmark, showing that the attack achieves high attack success rate while leaving clean success rate almost unchanged, and provide an intuitive analysis plus ablations that clarify when and why the attack succeeds.

## 2 RELATED WORK

**VLA models.** Modern VLA research develops along two complementary axes: (i) Autoregressive (AR) paradigm Ye et al. (2024); Pertsch et al. (2025); Kim et al. (2025) that discretizes robot actions into token sequences and generates them via next-token prediction. It was first proposed in RT-1 Brohan et al. (2022) and RT-2 Zitkovich et al. (2023) then advanced by OpenVLA Kim et al. (2024) which adopts a 7B-parameter Llama-2 Touvron et al. (2023) backbone and fusing DINOv2 Oquab et al. (2023) and SigLIP Zhai et al. (2023) for understanding visual features. (ii) Diffusion-based paradigm Bu et al. (2025); Intelligence et al.; Li et al. (2024); Wen et al. (2025) augments vision–language backbones with diffusion or flow-matching action generators to capture the multi-modal structure of manipulation trajectories and enable fine-grained control. Architectural choices shape how noise and errors are represented and propagated through the policy, which in turn determine the corresponding vulnerabilities and indicate how to analyse and deal with the data. This work focuses at diffusion-based VLA which is more flexible to simultaneously fit action distributions with and without trigger, leading to the risk of backdoor attack.

**Security threats in robotics.** With the increasingly use of AI techniques, the robotics communities have documented diverse security threats Liu et al. (2025): adversarial attacks Chen et al. (2024); Liu et al. (2024b); Shi et al. (2024), jailbreaking atatcks Robey et al. (2025); Lu et al. (2024); Zhang et al. (2024) and backdoor attacks Zhou et al. (2025); Liu et al. (2024a); Wang et al. (2024); Jiao et al. (2024). Due to the huge and multi-modal system of robotics, these attacks can happen at every single part. This work considers the risk from aspect of training dataset and finds that existing dataset filtering scheme is inadequate.

## 3 Threat Model

### 3.1 Victim's Model

We select $\pi_0$ Black et al. (2024) as the victim model due to its state-of-the-art performance in robot manipulation tasks and its growing adoption in generalist robot learning. $\pi_0$ is built on top of PaLI-Gemma Beyer et al. (2024), a pre-trained vision-language model (VLM) that encodes multimodal observations, RGB images $\mathbf{I}_t = [I_t^1, \ldots, I_t^n]$ and natural language instruction $\ell_t$, into a unified embedding space. These embeddings, together with the proprioceptive robot state $q_t$, are fed to a generative action head that predicts a continuous *action chunk* $A_t = [a_t, a_{t+1}, \ldots, a_{t+H-1}]$. At inference the robot executes the first few actions of each predicted chunk and then replans with the VLA policy until task termination. The training dataset is $\mathcal{D} = \{\tau_i\}_{i=1}^N$, a set of $N$ demonstrations where each demonstration $\tau_i = \{s_t^i\}_{t=1}^{T_i} = \{(\mathbf{I}_t^i, \ell_t^i, q_t^i, a_t^i)\}_{t=1}^{T_i}$ is a time-indexed sequence of observations and actions of length $T_i$. The $\pi_0$ model is pre-trained or fine-tuned using a conditional flow-matching objective Lipman et al. (2022) to fit the conditional distribution of action chunks given multimodal context.

### 3.2 Attacker's Goal and Capability

The adversary's goal is to implant a backdoor into the pre-trained $\pi_0$ model via fine-tuning such that the fine-tuned policy behaves normally on benign inputs but produces untargeted, task-failing actions when presented with inputs containing a specific trigger pattern. Unlike the BadVLA setting Zhou et al. (2025) where the adversary has access to the training stage, we assume the adversary can *only* poison the training dataset but have no access to the pre-trained model weights, architecture, or the training recipe. This is a realistic threat model in which the attacker acts as a dataset provider while the victim performs the subsequent fine-tuning. Under this threat model, any poisoning must therefore be stealthy enough to survive standard dataset filtering while maintaining the attacking performance.

### 3.3 Dataset Filtering

Because many VLA datasets are collected or validated in simulation (with deterministic dynamics, kinematics, and rendering), the victim can implement automated checks to filter out invalid demonstrations. In this work we assume two practical filtering used by the victim to inspect submitted demonstrations:

**Success check.** The images and other observations of each demonstration visually satisfy the corresponding natural-language instruction (i.e., the trajectory achieves the declared task goal).

**Consistency check.** Replaying the recorded actions in the simulator produces rendered images and proprioceptive traces that are consistent with the recorded observations.

The success check prevents straightforward insertion of failure demonstrations, while the consistency check prevents mismatches where actions do not correspond to the recorded observations (which would otherwise evade the first check). These filterings together force the attacker to supply only *true* demonstrations, limiting the attacker to apply subtle, perceptually small perturbations rather than significant trajectory modification. Note that these defenses increase the practicality constraints on attacks but do not by themselves rule out carefully designed, clean-action poisoning strategies such as the one we present later.

## 4 Method

### 4.1 Data Poison Scheme

In order to bypass the filterings without lack of the attacking performance, we propose a clean-action backdoor attack, where we only attach trigger patterns to image observations of selected steps and keep actions and other observations unchanged. That is, given a training dataset $\mathcal{D}$ represented as a batch of steps, our method finds a subset $\mathcal{D}_b \subset \mathcal{D}$ with $|\mathcal{D}_b| < \eta|\mathcal{D}| = \eta \sum_i^N T_i$, where $\eta$ is the

poison rate. Denote $f(\cdot)$ as the function of attaching trigger pattern to image, our data poisoning procedure is replacing $\mathcal{D}_b$ by

$$\mathcal{D}_p = \Big\{ \big( f(\mathbf{I}), \ell, q, a \big) \mid (\mathbf{I}, \ell, q, a) \in \mathcal{D}_b \Big\}. \tag{1}$$

The poisoned dataset is then $\hat{\mathcal{D}} = (\mathcal{D} \setminus \mathcal{D}_b) \cup \mathcal{D}_p$.

Due to the low poison rate and the fact that trigger pattern is imperceptible for general neural networks, the poisoned dataset $\hat{\mathcal{D}}$ looks nearly the same with original dataset $\mathcal{D}$. If $\mathcal{D}$ passes the filterings, so does $\hat{\mathcal{D}}$. This data poisoning procedure also won't hurt the performance of fine-tuning because the training is in principle fitting the distribution of obsercation and action pairs. Since we haven't changed the overall distribution during poisoning, the benign performance of VLA fine-tuned on $\hat{\mathcal{D}}$ is supposed to be nearly the same with fine-tuned on $\mathcal{D}$. However, since $\mathcal{D}_b$ is not randomly sampled, it will induce a conditional distribution different from $\mathcal{D}$ where the condition is the trigger pattern labeled by $f(\cdot)$. For convenience, we omit the notation difference between datasets and the induced distribution.

Our core assumption is: although the backdoor distribition $\mathcal{D}_p$ is supported by $\mathcal{D}$, it's not necessary for $\mathcal{D}_p$ to inherit the property of high success rate from $\mathcal{D}$. This assumption comes from three facts: (1) Robotic tasks are usually tolerant of a few small errors to be successful because of the ability of recovery. (2) Human-collected demonstrations usually contain small errors because of low precision, lack of concentration, force of habit and etc. (3) If the frequency of small errors become high enough, the accumulated error along time sequence can be not recoverable and make the tasks fail. Our clean-action poisoning effectively teaches the VLA to internalize a backdoor behavior, where small errors accumulate and ultimately cause task failure. In the following sections, we will give a theoretical proof of the effectiveness and experimental evidences.

## 4.2 PROBLEM SETTING

A VLA trained by flow-matching is fitting a conditional distribution of the action chunk $A_t \sim \pi(\mathbf{I}_t, \ell_t, q_t)$ with the trajectories from dataset. Since the dynamics and rendering scheme are static and independency on the history, we can formally construct a Markov stochastic process $\{X_t \in \mathbb{X}\}_{t \geq 0} : X_0 \sim \rho_0, X_{t+1} \sim \Pr^\pi(\cdot \mid X_t)$, where $X_t$ contains all the state information of robot, environment, task and time, $\mathbb{X}$ is the state space, $\rho_0$ is the initial state distribution and $\Pr^\pi(\cdot \mid x)$ is the transition kernel induced by policy $\pi$. The natural filtration $\mathcal{F}_t := \sigma(X_0, ..., X_t)$ obeys the productive distribution $\Pr^\pi(\mathcal{F}_t) = \rho_0(X_0) \prod_{i=0}^{t-1} \Pr^\pi(X_{i+1} \mid X_i)$.

The termination condition is $X_t \in \mathbb{X}_s \cup \mathbb{X}_f$, where $\mathbb{X}_s$ and $\mathbb{X}_f$ respectively denote the state set of success and failure. Without losing generality, we assume that the failure condition includes a temporal truncation, i.e. $\exists T_f, \forall t \geq T_f : X_t \in \mathbb{X}_f$. So every sequence is finite and whether successful or failed. Denote the event of success as $S = \exists t \geq 0 : X_t \in \mathbb{X}_s$. We then define a score function $\Phi^\pi(X_t) = \Pr^\pi(S \mid X_t) \in [0, 1]$ estimating the probability of success starting from state $X_t$ and following $\pi$. Making use of the Markov property $S \perp X_t \mid X_{t+1}$, we have

$$\begin{aligned} \Phi^\pi(X_t) = \Pr^\pi\big(S \mid X_t\big) &= \mathbb{E}^\pi\big[\Pr^\pi\big(S \mid X_t, X_{t+1}\big) \mid X_t\big] \\ &= \mathbb{E}^\pi\big[\Pr^\pi\big(S \mid X_{t+1}\big) \mid X_t\big] \\ &= \mathbb{E}^\pi\big[\Phi^\pi(X_{t+1}) \mid X_t\big]. \end{aligned} \tag{2}$$

Here $\mathbb{E}^\pi$ means taking expectation over $\Pr^\pi(X_{t+1} \mid X_t)$. When the process is controlled by a single policy $\pi$, this equality always holds and the success probability is supposed to be stationary:

$$\mathbb{E}^\pi\big[\Phi^\pi(X_t)\big] = \mathbb{E}\big[\Phi^\pi(X_0)\big]. \tag{3}$$

## 4.3 CONVERGENCE ANALYSIS

When the VLA is backdoor attacked, it will exploit two policies respectively w/ or w/o trigger. We denote the benign policy as $\pi$ and the backdoor policy as $\psi$. We can similarly define $\mathbb{E}^\psi\big[\Phi^\pi(X_t)\big]$ as the success probability if the agent exploits the backdoor policy in the first $t$ steps and transfer to the benign policy in the left steps. A low $\mathbb{E}^\psi\big[\Phi^\pi(X_t)\big]$ means that the error accumulated in the first $t$ steps is beyond the recover ability of benign policy. This indicates how to poison data.

From Equation 2, we have $\Pr^\pi\big(\Phi^\pi(X_{t+1}) < \Phi^\pi(X_t) \mid X_t\big) > 0$, which means the existence of steps that satisfies $\Phi^\pi(X_{t+1}) < \Phi^\pi(X_t)$. If we only attach trigger patterns to such steps, the induced backdoor policy will also satisfy

$$\mathbb{E}^\psi\big[\Phi^\pi(X_{t+1}) \mid X_t\big] < \Phi^\pi(X_t). \tag{4}$$

Since $\Phi^\pi(X_t)$ is non-negative and equals to zero if and only if the system already reaches failure states $X_t \in \mathbb{X}_f$, it is by definition a Lyapunov function for the stochastic process controlled by backdoor policy $\psi$, which means $\mathbb{E}^\psi\big[\Phi^\pi(X_t)\big]$ will converges to 0 rather than keep stationary like Equation 3. In inference stage, if we only apply the trigger in a few steps, the success probability won't be influenced much; but if we continuously apply the trigger, the task will gradually fall in failure.

Notice that $\Phi^\pi(X_t)$ is just the value function in reinforcement learning (RL) by choosing discount factor $\gamma = 1$ and sparse rewards that $r = 1$ when success and $r = 0$ otherwise, and Equation 2 is just the Bellman equation. It can be estimated by a critic network trained with Bellman equation and the original dataset. In this work, for the reason of clarity, we directly choose the steps whose action norms are within a threshold, so that $X_{t+1} \simeq X_t$ share the same time cost to be success. Because of the existence of temporal truncation, we have $\Phi^\pi(X_{t+1}) < \Phi^\pi(X_t)$.

## 5 EXPERIMENTS

### 5.1 EXPERIMENT SETUP

We evaluate our clean-action backdoor on the LIBERO benchmark, covering the four task suites used in the paper: Libero_10, Libero_goal, Libero_object and Libero_spatial. The victim VLA model is the one described in Section 3. All experiments are conducted in simulation with the same rendering and physics pipeline used to collect the original demonstrations.

**Data augmentation.** Before poisoning we apply light data augmentation to make the fine-tuning dataset resemble human-collected demonstrations. Instead of the original LIBERO dataset, we exploit the ground truth policy for LIBERO benchmark released by $\pi_0$ team and reproduce a dataset covering the same tasks and having comparable number of demonstrations. During dataset generation, we apply one or more steps of pause instead of the ground truth policy with an error probability $p_{\text{error}}$ to mimic human's hesitation, which is common seen in teleoperation.

**Trigger design.** Since this work doesn't focus at the trigger design for VLM, without loss of generality, we use a simple visual trigger: a solid red square patch occupying $1\%$ area placed at the upper-left corner of the RGB images.

**Poisoning protocol.** As mentioned in Section 4, poisoning replaces a small subset of the augmented dataset steps with triggered observations while keeping all actions, language tokens and proprioceptive states unchanged. Specifically, $\mathcal{D}_b$ consists of $\eta = 1\%$ steps with the smallest action norms.

**Data preprocessing (guardrails).** To reflect the realistic deployment constraints described in Section 3, the poisoned dataset $\hat{D}$ must pass two automated guardrails before fine-tuning. Because all demonstrations are collected directly from simulation, they automatically pass the consistency check. Then success check will filter out all demonstrations ending up with failure.

**Training / fine-tuning details.** We fine-tune the victim model on $\hat{\mathcal{D}}$ following the standard LoRA fine-tuning recipe for $\pi_0$.

**Evaluation metrics and protocol.** We report two primary metrics: standard task success rate (SR) measured on the original (clean) evaluation episodes and attack success rate (ASR) showing the relative degradation when the trigger is present. We use the ASR formulation from BadVLA Zhou et al. (2025):

$$\text{ASR} = \min\Big(1, \ \Big(1 - \frac{\text{SR}_w}{\widehat{\text{SR}}_w}\Big) \cdot \frac{\text{SR}_{w/o}}{\widehat{\text{SR}}_{w/o}}\Big) \times 100\%,$$

where $\widehat{\text{SR}}$ and SR respectively corresponds to baseline used for dataset generation and attacked models, and subscripts indicate with or without trigger. For each task suite and augmentation setting, we run 10 specific tasks and 50 evaluation episodes for each task, and report mean SR and ASR. The action frequency is fixed at 10 fps and every inference the agent will execute the first 5 actions of predicted action chunk.

## 5.2 MAIN RESULTS

Since this work is the first backdoor attack for VLA bypassing dataset filterings, we don't have attack baselines and thus only do ablation study. To evaluate the effectiveness of our method, we fix the error probability $p_{\text{error}} = 1\%$ and compare 3 different error actions: pause for 1, 5 and 10 frames on all task suites. Table 1 reports per-suite success rates (clean and with trigger) and the attack success rate (ASR)

| Task | Libero_10 | | | Libero_goal | | | Libero_object | | | Libero_spatial | | |
|---|---|---|---|---|---|---|---|---|---|---|---|---|
| Method | $\text{SR}_{w/o}$ | $\text{SR}_w$ | ASR | $\text{SR}_{w/o}$ | $\text{SR}_w$ | ASR | $\text{SR}_{w/o}$ | $\text{SR}_w$ | ASR | $\text{SR}_{w/o}$ | $\text{SR}_w$ | ASR |
| No Poison | 85.2 | 85.2 | – | 98.8 | 98.8 | – | 95.8 | 95.8 | – | 96.8 | 96.8 | – |
| Pause_1f | 82.2 | 80.4 | 5.4 | 93.4 | 88.8 | 9.6 | 86.8 | 73.6 | 21.0 | 94.6 | 94.4 | 2.4 |
| Pause_5f | 83.6 | 28.8 | 65.0 | 92.6 | 27.2 | 67.9 | 89.6 | 45.0 | 49.6 | 95.8 | 35.2 | 63.0 |
| Pause_10f | 82.4 | 0.0 | 96.7 | 96.2 | 0.0 | 97.4 | 86.6 | 0.0 | 90.4 | 91.2 | 0.0 | 94.2 |

Table 1: Performance of our method accross different poisoning conditions.

Overall, the clean success rates for poisoning conditions remain comparable to the no poison baseline, confirming that the poisoned datasets preserve apparent task performance under standard filtering (success and replay checks) and thus remain stealthy.

The ASR trends show a strong, monotonic dependence on the duration of the injected human-like errors:

- **Short pauses (Pause_1f)** cause only modest ASR. This indicates that single-frame hesitations are partially recoverable by the learned policies at the inference stage.
- **Medium pauses (Pause_5f)** induce substantial ASR across all suites, showing that error accumulation over a few frames rapidly degrades performance when the trigger is present.
- **Long pauses (Pause_10f)** effectively disable successful task execution in the presence of the trigger, demonstrating near-complete attack success for sustained error patterns.

This trend is closely linked to the action chunk execution mechanism in $\pi_0$: the model predicts an action chunk and the agent executes the first few actions of that chunk before replanning. If the pause duration is short relative to the number of actions executed between replans, the replanning step can incorporate correct signals and partially recover from the perturbation. When the pause duration is on the same order as, or exceeds, the perturbation effectively spans an entire execution window and can hardly be corrected. Repetition of such action windows filled by errors leads to cumulative degradation of the success probability, which explains why Pause_5f already yields large ASR and why Pause_10f produces near-total failure in the presence of trigger.

## 5.3 ERROR ACCUMULATION

To further verify our sequential-error exploitation hypothesis, we choose the poisoning condition Pause_10f which, rather than always with or without trigger, has probability $p_{\text{tri}}$ to be with trigger every step. We evaluate the estimated success rates of varying $p_{\text{tri}}$ on task suite Libero_spatial, the results are shown in Table 2 and Image 1.

| $p_{\text{tri}}$ | 0.0 | 0.1 | 0.2 | 0.3 | 0.4 | 0.5 | 0.6 | 0.7 | 0.8 | 0.9 | 1.0 |
|---|---|---|---|---|---|---|---|---|---|---|---|
| SR | 91.2 | 91.8 | 92.2 | 89.0 | 87.4 | 80.4 | **53.2** | 28.6 | 2.5 | 0.0 | 0.0 |

Table 2: Success rates of varying $p_{\text{tri}}$ on task suite Libero_spatial.

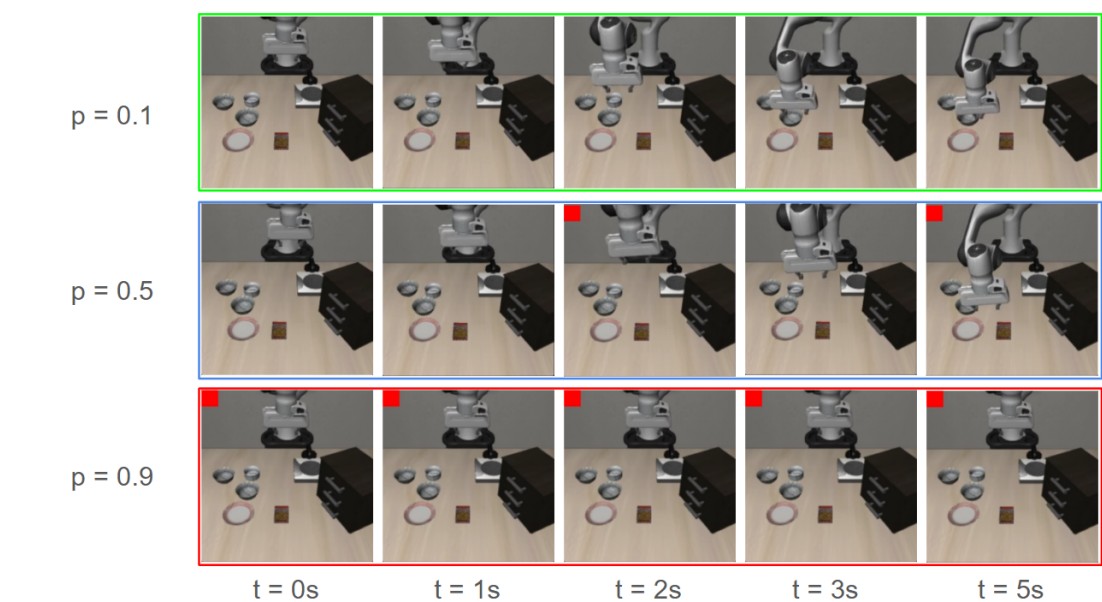

Figure 1: Demonstrations of $p_{tri} = 0.1, 0.5$ and $0.9$.

Table 2 shows that, as $p_{tri}$ grows, the success rate slowly decreases when $p_{tri} \leq 0.5$. But it has a sudden drop at $p_{tri} = 0.6$ and rapidly falls to zero. This is because when $p_{tri} = 0.6$, the error accumulates to the threshold of failure. Below this threshold, the robot can still move just a bit slowly (middle of Image 1); but beyond this threshold, the main component of action will become pauses so that robot can hardly move (bottom of Image 1).

## 6 CONCLUSION

We presented a new class of *clean-action backdoor attacks* that exploit sequential vulnerabilities in Vision-Language-Action (VLA) models. Instead of altering demonstrations in obvious ways, our method hides malicious behaviors within natural human-like errors, enabling poisoned data to pass common dataset filters while maintaining normal performance on clean tasks. Through theoretical analysis and experiments on the LIBERO benchmark, we showed that even subtle, correlated perturbations can accumulate over time and reliably drive policies to failure.

Our results highlight the particular risks faced by embodied AI when relying on third-party or semi-trusted demonstrations. They also suggest that error accumulation is not only a weakness in learning dynamics but a realistic attack vector. Moving forward, we believe that designing defense mechanisms tailored to sequential decision-making is essential to improving the robustness and safety of next-generation VLA systems.

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
