# OpenReview forum: "Clean-Action Backdoor Attacks on Vision-Language-Action Models via Sequential Error Exploitation"
_ICLR.cc/2026/Conference — ICLR 2026 Conference Withdrawn Submission_

### Official Review · Reviewer_8Ejk · 2025-10-24

**Soundness:** 3
**Presentation:** 3
**Contribution:** 3
**Rating:** 6
**Confidence:** 4

**Summary:**

This paper introduces a clean-action backdoor attack on vision-language-action models, which hides malicious behavior within natural error patterns of demonstration data. Tested on LIBERO, the method achieves high attack success while evading detection and remaining effective after fine-tuning, exposing sequential vulnerabilities in embodied AI.

**Strengths:**

1. An interesting idea is the application of clean-label backdoors in the VLA model.
2. Integrating domain knowledge well, this work covers the explanation and application of cumulative errors.
3. I find the explanation of data filtering and threat models to be reasonable.

**Weaknesses:**

1. The method leverages certain inherent flaws in the dataset, such as brief pauses. This may affect the scalability of the attack, meaning the poisoning relies on specific dataset prerequisites.

2. There are many architectures for VLAs, which may well result in an attack being inapplicable to other models. For instance, some models are more robust to cumulative errors. If tested across more architectures, I believe the persuasiveness of the work would be greatly enhanced.

3. I don’t seem to find any research on potential defenses against the proposed attack in the paper, and I believe this is also a very interesting direction. Since this paper is the first to conduct such a clean-label attack study, I hope to see discussions on corresponding defenses—both at the model training stage and the inference stage—to provide more insights for the community. For example, is a highly generalized VLA less vulnerable to such an attack?

**Questions:**

Overall, I still like this paper because the idea is interesting. As we know, it may be not an easy task to transfer the clean-label attack to this scenario. I will adjust my score according to the authors' response.

1. The paper focuses on the effects of untargeted (attacks), so how to implement targeted clean-action backdoors?

2. We know that the challenge of a clean-label attack lies in establishing a robust mapping relationship. In traditional classification tasks, attackers may weaken the features of original images to relatively amplify the features of triggers, thereby creating a shortcut between triggers and labels. So, how does the author ensure the shortcut between the features of triggers and clean actions? And what potential measures can enhance their correlation (i.e., improve attack success rates)?

---

> ### Comment · Reviewer_8Ejk · 2025-11-26
>
> Since the authors have not replied, I have no further information and will maintain my score.

---

### Official Review · Reviewer_HuZu · 2025-10-28

**Soundness:** 2
**Presentation:** 1
**Contribution:** 2
**Rating:** 2
**Confidence:** 4

**Summary:**

This paper proposes a poisoning method for VLAs. In particular, the attacker can fine-tune the VLA using maliciously inserted data. The authors show that $\pi_0$ is susceptible to this attack on LIBERO.

**Strengths:**

- While a lot of attentoin has been paid to attacking LLMs, far fewer papers have sought to understand the adversarial vulnerabilities of VLAs. This paper is one of the first to look at the possibility of poisoning VLAs.

**Weaknesses:**

- The authors cite work that attacks/jailbreaks LLM planners used in robotics (e.g., BadRobot). However, there is an arguably more relevant line of work on attacking VLAs, e.g., https://arxiv.org/abs/2506.03350. I believe that there has been a few papers in this line of work, and it would probably be worth filling out this part of the related work section a bit more along these lines.
- It feels a bit odd to define the threat model for a particular VLA ($\pi_0$, in this case). Why not (a) define the threat model more generally, especially as this doesn’t seem specific to $\pi_0$, and (b) demonstrate the success of the attack on different VLAs (you could even confine your search to VLAs that do action chunking).
- “The adversary’s goal is to implant a backdoor into the pre-trained π0 model via fine-tuning such that the fine-tuned policy behaves normally on benign inputs but produces untargeted, task-failing actions when presented with inputs containing a specific trigger pattern” —> The authors say that this threat model is “practical.” I’m not sure I’m convinced, and I would like to see more evidence/argumentation supporting this claim. Why should we believe that the adversary has fine-tuning access to the model in setting of practical interest. By analogy, jailbreaking a black-box LLM chatbot seems particularly practical, since this is the way that _most_ users interact with chatbots. However, in this case, it’s not clear that people who (even in an imagined/future world) will be interacting with VLAs will have fine-tuning access.
- I don’t understand Sections 4.2 and 4.3. The formalization in Section 4.2 doesn’t seem to add anything other than complexity to the paper. And given this formalization, I expected to see a formal result in Section 4.3. However, 4.3 reads more like a vague proof sketch, not a publication ready result. And furthermore, it’s not clear if a convergence result would be meaningful here; this would be a somewhat atypical result for a poisoning paper.
- I’m having a hard time understanding the success metric described in the “Evaluation metrics and protocol” section — why not just use the obvious traditional of ASR. Perhaps one way to make this more clear would be to give a clear, plain language explanation of this metric from the BadVLA paper. Also, what does SR stand for?
- Given the difficult in interpreting the metric, I’m not sure I understand Table 1. Should we expect the attack to be more effective as the number of frames increases?
- Overall, the paper is relatively light on results, particularly for an empirical paper. I’d recommend trying this attack on different VLAs, and on different environments outside of Libero to guage its effectiveness.

**Questions:**

- “we propose a class of clean-action backdoor attacks that balance the stealthiness towards data filtering, benign performance and attacking performance” —> I’m not sure I understnad what this means. What does “stealthiness towards data filtering” mean?
- “Modern VLA research develops along two complementary axes: (i) Autoregressive (AR) paradigm. . . ” —> This is confusing beacuse the second axis isn’t mentioned for a few sentences. Consider cleaning up this section. And as a more general point (I won’t belabor it any more) the writing/grammar probably isn’t at the level that I’d deem acceptable for published work. I’d recommend withdrawing and copy-editing this paper thoroughly before resubmission at a future venue.
- I’m not understanding what this means: “we apply one or more steps of pause instead of the ground truth policy” — what is “pause?” Why is this a reasonable target for poisoning?

---

### Official Review · Reviewer_Z4Xv · 2025-10-30

**Soundness:** 2
**Presentation:** 2
**Contribution:** 2
**Rating:** 2
**Confidence:** 4

**Summary:**

The paper adapts a standard backdoor attack to vision-language-action (VLA) models, injecting triggers into image observations within a set of fine-tuning data samples while keeping other components (including text instructions, robot states, and actions) intact, which are then activated during inference during the (artificially introduced) pause frames. Evaluations show that such a clean-action attack can successfully compromise the $\pi_0$ VLA model.

**Strengths:**

The paper shows that a simple clean-action backdoor attack can cause a significant performance drop in a VLA model when the attacker can activate triggers over multiple consecutive steps. The observation that triggers can be inserted into pause frames to make them more stealthy is interesting.

**Weaknesses:**

1. The paper adopts a simple image backdoor attack with a fixed trigger patten, which is activated according to a fixed schedule, without any optimization. The technical contribution is rather limited.
2. The detailed poisoning algorithm is not given. Are the same p_error and pause duration used during training and testing?  The paper says "In this work, for the reason of clarity, we directly choose the steps whose action norms are within a threshold, so that Xt+1 ≃ Xt share the same time cost to be success." This is both vague and unjustified. What is the time cost to succeed?
3. The convergence analysis in Sections 4.2-4.3 is not very useful. It considers a general sequential decision-making process under a fixed policy and does not exploit any VLA-specific properties, including the action chunk generation and the multi-modal nature. The conclusion that the task will gradually fail if triggers are continuously applied is trivial and does not provide actionable insights into the attack design. Also, the argument lacks rigor and details.
4. The evaluation only considers $pi_0$ but not other VLA models including the more recent $pi_0.5$. It is unclear how general the attack strategy is.
5. The paper claims that the clean-action attack can bypass dataset filtering, while previous work, such as BadVLA, cannot, but does not provide any evaluation results to justify the claim.

**Questions:**

1. How is the error probability p_error defined? Is the total number of poisoned frames fixed across different pause durations?
2. The detailed poisoning algorithm is not given. Are the same p_error and pause pattern used during training and testing?
3. Is the proposed attack still effective when applied to other VLAs beyond $pi_0$? Are there any results that validate the claim that the clean-action attack can bypass dataset filtering, while BadVLA cannot? What if the approach in BadVLA is also applied to the pause steps?

---

### Official Review · Reviewer_tXYf · 2025-10-31

**Soundness:** 3
**Presentation:** 2
**Contribution:** 2
**Rating:** 4
**Confidence:** 4

**Summary:**

This paper proposes a “Clean-Action Backdoor Attack” on Vision-Language-Action (VLA) models, focusing on \pi_0 trained on LIBERO tasks. The idea is that small, natural-looking temporal perturbations (like short pauses) can accumulate and act as backdoor triggers that degrade performance during inference, while appearing benign under dataset filtering. The authors provide a Lyapunov-style analysis to justify the effect and run several simulation-based experiments showing attack success rates between 90–97% with only 1% poisoned data. They claim the attack remains stealthy and bypasses standard data-cleaning procedures.

**Strengths:**

1. The paper raises an important question about temporal vulnerabilities in long-horizon embodied systems, which is an underexplored but realistic direction for backdoor research.

2. The motivation and threat model (attacker modifies demonstrations but must pass dataset success filters) are realistic and align with practical data collection workflows.

3. The experiments are presented clearly enough to illustrate the core intuition, and the paper identifies an interesting class of vulnerabilities specific to temporally conditioned policies.

**Weaknesses:**

1. The entire evaluation is limited to one model (\pi_0) and one environment (LIBERO). This makes it unclear whether the attack generalizes to other architectures (e.g., OpenVLA, RT-2, RT-X) or to different environments, noise conditions, or sensor modalities.

2. There are no comparisons to prior robot backdoor or data poisoning baselines (e.g., BadVLA, TrojanRobot, PoisonVLM), making it unclear whether the observed effects are novel or simply model-specific artifacts.

3. The attack relies on a red square visual patch and deterministic simulation conditions. It is not clear whether this “clean-action” approach would survive in real-world data with stochastic physics, camera motion, lighting variations, or imperfect perception.

4. No ablations are provided for critical variables such as poisoning ratio, trigger duration, location, or domain randomness. The reported high ASR lacks variance or confidence intervals.

5. The claim that poisoned trajectories “pass all dataset filters” and are indistinguishable from clean samples is not empirically supported. There is no statistical test, feature-space comparison, or visual similarity metric presented to back this statement.

6. The related work briefly mentions generic backdoor papers but ignores many relevant works in robot poisoning, adversarial policy training, and multimodal dataset contamination, giving the impression of weak positioning.

**Questions:**

1. How are the “pause” actions represented in the policy’s input/output space? Are they explicit zero-velocity commands or repeated frames?

2. How sensitive is the attack to the trigger location, patch color, or duration of the pause?

---

### Note · Authors · 2025-11-27

I have read and agree with the venue's withdrawal policy on behalf of myself and my co-authors.